# 1.2 kW, 20 kHz Nanosecond Nd:YAG Slab Laser System

**Hao Liu** [1,2], **Jisi Qiu** [1], **Yanzhong Chen** [1], **Haocheng Wang** [1], **Tianqi Wang** [1], **Yueliang Liu** [1], **Xiaoquan Song** [3] and **Zhongwei Fan** [2,*]

1   Aerospace Information Research Institute, Chinese Academy of Sciences, Beijing 100094, China; liuhao@aircas.ac.cn (H.L.)
2   University of Chinese Academy of Sciences, Beijing 101408, China
3   Beijing Institute of Tracking and Telecommunication Technology, Beijing 100094, China
*   Correspondence: fanzhongwei@ucas.ac.cn

**Abstract:** In this paper, we develop a kW-level high-repetition-rate nanosecond master oscillator power amplifier (MOPA) laser system, employing a structure of fiber, Nd:YVO$_4$, and Nd:YAG hybrid amplification. A tunable fiber seed source is used for adjustable pulse repetition frequency and pulse width. The Nd:YVO$_4$ pre-amplifier, which is dual-end-pumped, achieves high gain while maintaining good beam quality, and the high-power side-pumped Nd:YAG slab main-amplifier enables efficient power amplification. The repetition rate of the output laser can be adjusted within the range of 1~20 kHz, and the pulse width can be tuned within the range of 10~300 ns. The seed output is 6 mW at a repetition frequency of 20 kHz; we achieve an average output power of 1240 W with a total power extraction efficiency of 39.1% and single-pulse energy of 62 mJ at a pulse width of 301 ns. This parameter-controllable high-power laser holds promise for applications in the laser cleaning of complex surface contaminants.

**Keywords:** high-power laser; diode-pumped; optical amplifiers; lasers pulsed

## 1. Introduction

Laser with high repetition rate ($\geq$10 kHz), high single-pulse energy ($\geq$10 mJ), and narrow pulse width have significant application value and demands in various fields, particularly within advanced manufacturing sectors such as laser cleaning, laser processing, and laser cutting [1–3]. Additionally, it possesses a unique advantage in high-repetition rate particle image velocimetry (PIV) [4–6].

High single-pulse energy laser systems at the joules to multi-joules level have been extensively reported. However, they typically operate under low repetition rate of less than 1 kHz, exhibiting complex structures and considerable physical dimensions. For example, in 2016, our group reported an average 336 W from an Nd:YAG MOPA nanosecond laser at the repetition rate of 100 Hz, with a far-field beam spot 1.71 times the diffraction limit [7]. In 2018, Zhijun Kang et al. reported a rod amplifier laser based on Nd:YAG, with a single-pulse output energy of 1 J and a repetition rate of 500 Hz [8]. On the other hand, a multitude of institutions have reported laser systems with an extremely high repetition rate but low single-pulse energies. Chunhua Wang et al. reported a 70 kHz, 90 ps laser pulse from a double-passing end-pumped Nd:YVO$_4$ rod amplifier laser, with the pulse energy of 143 μJ [9]. In 2016, they utilized a microchip laser seed source with a pulse width of 95 ps, which was amplified by two stages of Nd:YVO$_4$ end-pumped amplifiers, resulting in an output with a repetition rate of 100 kHz and a single-pulse energy of 320 μJ [10]. Fu, X. et al. demonstrated a four-stage Nd:YVO$_4$ amplifier MOPA laser, with a repetition rate of 500 kHz and a single-pulse energy of 240 μJ [11]. F. Saltarelli et al. demonstrated a 350 W with 940 fs pulses at the output of an Yb:YAG thin-disk oscillator mode-locked with a semiconductor saturable absorber mirror (SESAM), with a repetition rate at 8.88 MHz and 40 μJ pulse energy [12]. Yiping Zhou et al. reported a 5 kHz sub-nanosecond master oscillator power



amplifier (MOPA) Nd:YVO$_4$ laser system with a single-pulse energy of 4.2 mJ [13]. Yongxi Gao demonstrated a 417 W, 175 kHz, 2.38 mJ innoslab chirped pulse amplification laser [14]. Additionally, fiber lasers have achieved high levels of average power output [15–17], but their single-pulse energy is limited by the issue of fiber core damage, resulting in relatively low single-pulse energy, typically in the range of a few millijoules. Furthermore, due to the small diameter of the fiber core, nonlinear effects such as stimulated Brillouin scattering (SBS), stimulated Raman scattering (SRS), and self-phase modulation (SPM) easily occur during the amplification process. These nonlinear effects pose significant limitations on the enhancement of peak power for narrow linewidth nanosecond pulse fiber lasers [18–21]. Consequently, the realization of high-repetition-rate, high-energy frequency-doubled lasers using fiber lasers becomes extremely difficult, imposing restrictions on their application in precision measurement domains.

For example, in laser-cleaning applications, when high-repetition-rate pulsed lasers interact with material surface substances, contaminants rapidly absorb laser energy, leading to a localized high temperature zone. Subsequent pulsed lasers further increase the temperature in this area. At this point, the contaminants on the surface undergo ablation and gasification, forming keyholes, which inhibit temperature diffusion. This results in an accelerated temperature rise within the keyholes, generating intense transient ablation and gasification effects and facilitating impurity removal. Clearly, a higher repetition rate and pulse energy lead to faster cleaning speeds. The pulse duration falls within the nanosecond scale, ensuring excellent cleaning effects while preventing damage to the substrate from thermal accumulation. Additionally, for composite contaminants, which typically have complex compositions, a single-laser parameter may not achieve optimal cleaning results. Consequently, it is necessary to adjust laser parameters such as repetition rate and pulse width flexibly to address the varying interaction mechanisms between different components and the laser.

To achieve a laser system with both high repetition rates and high single-pulse energy output, as well as flexible temporal control of the pulse waveform, we have developed a Nd:YAG slab hybrid amplifier based on MOPA configuration with a fiber seed source. The pulse width and repetition rate of laser system are determined by the characteristics of the seed source, which offers the advantage of tunability in both pulse width and repetition rate. This design allows for versatile applications in fields such as laser cleaning and PIV diagnostics, addressing the limitations of traditional solid-state/fiber lasers.

In this configuration, the seed source comprises a fiber laser that provides a low-power signal light in the milliwatt range. The pulse waveform is dynamically controlled through the utilization of an acousto-optic modulator (AOM). For pre-amplification, a four-stage double-end-pumped Nd:YVO$_4$ laser amplifier is employed, capitalizing on the advantageous characteristics of the large stimulated emission cross-section exhibited by Nd:YVO$_4$ crystals. This design enables a substantial amplification of the seed light, thereby enhancing the overall energy efficiency of the laser system while ensuring exceptional beam quality. In the main amplification stage, a two-stage high-power continuous-wave (CW) side-pumping scheme is implemented using a parallelogram-shaped Nd:YAG slab laser amplifier. By capitalizing on the high energy storage capability inherent in the slab amplifier, the laser system achieves a remarkable output of high-power laser amplification. In this work, 1240 W of average power is obtained, corresponding to the single-pulse energy of 62 mJ at the pulse width of 300 ns. Furthermore, the pre-compensation technique for the seed pulse waveform enabled flexible control of the laser temporal profile, making it highly promising for applications in laser cleaning.

## 2. Experimental Setup

The experimental setup of the 1.2 kW, 20 kHz, 300 ns MOPA laser system is shown schematically in Figure 1; the laser seed source consists of a butterfly-packaged distributed feedback (DFB) laser diode, a fiber acousto-optic modulator, and two stages of Yb-doped fiber amplifiers. This configuration provides a highly stable seed laser output with ad-

justable repetition rate (1~20 kHz) and pulse width (10~300 ns). The average power of the seed is 6 mW at a repetition rate of 20 kHz. The seed laser is collimated by lens L1 and then enters a beam splitting and isolation protection unit consisting of a half-wave plate, Faraday rotator (FR), and polarizing beam splitter (PBS). It is then focused by lens L2 and directed into the LD dual-end-pumped pre-amplification module PA-1. This configuration optimizes the mode matching between the signal beam and the pump beam distribution within the gain medium. The HR-coated flat mirror M1 reflects the amplified laser beam. It goes back into the PA-l for the second-passing extraction. This process achieves significant gain amplification of the weak seed signal laser while maintaining high beam quality. After that, the polarization of the main beam is modified using a half-wave plate and a Faraday rotator. This allows it to pass through the PBS and enter the beam combining unit consisting of L3 and L4. It then goes through the pre-amplification modules PA-2, PA-3, and PA-4 in sequence. After three stages of unidirectional amplification, the output power reaches around ~100 W, providing enough pre-amplification power for the subsequent main power amplification stage. After the pre-amplification stage, the beam passes through a 4F spatial filter unit composed of L5, L6, and SF1 for spatial shaping and filtering. The beam is expanded with a ratio of 1:4. It is then further expanded and collimated in the y-direction with a 1:4 beam expansion using the lens group L7 and L8 before entering the main amplification stage. The main amplification stage consists of two side-pumped Nd:YAG slab laser amplification modules, A-1 and A-2. The beam is incident at the Brewster angle and undergoes multiple total internal reflections (TIR) inside the slab, propagating in a zigzag pattern. After being reflected by angle mirrors, the beam passes through the slab crystal for the second time with different incident directions, eliminating the gain blind spots present in the zigzag path and extracting the stored energy within the amplification module more effectively. By combining uniform pumping light and efficient thermal management design, high-efficiency power extraction and amplification are achieved, resulting in kilowatt-level output power.

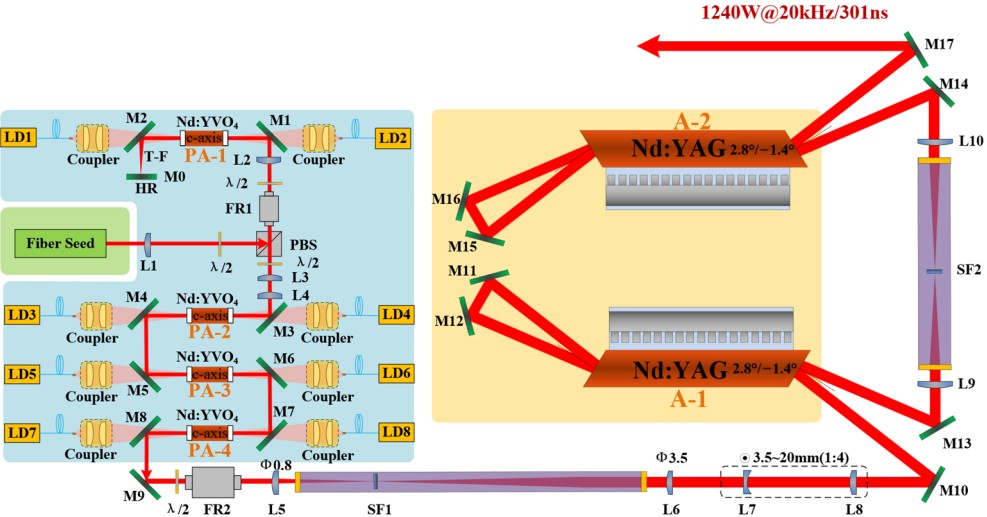

**Figure 1.** Experimental setup of laser system.

### 2.1. Pre-Amplifier Setup

The pre-amplification unit consists a four-stage end-pumped Nd:YVO$_4$ laser amplifier. The 0.3 at.% doped a-cut composite Nd:YVO$_4$ crystal (provided by Crylaser Inc., Chengdu, China) with dimensions of 4 mm × 4 mm × 20 mm is used in the amplifier as the gain medium. Nd:YVO$_4$ is an efficient four-level laser gain medium with a pump light centered around 808 nm and an output laser centered at 1064 nm. Compared to the Nd:YAG crystal, the Nd:YVO$_4$ crystal exhibits a stimulated emission cross-section approximately five times larger. This implies that the Nd:YVO$_4$ crystal can achieve a higher gain factor under the same operating conditions, making it an ideal material for small-signal amplification. Both

end faces of the Nd:YVO$_4$ crystal are anti-reflection (AR), coated at 808 nm and 1064 nm, with transmittance rates of 99.9% and 98%, respectively. The crystal is encased in a 0.1 mm thick indium foil and bonded to a copper heat sink, which is cooled by circulating water at a constant temperature of 20 °C. M1~M8 are AR coated at 808 nm and HR coated at 1064 nm at an incidence angle of 45°.

Due to the relatively low output power of 6 mW from the laser seed source, it falls within the regime of a small signal for amplifiers. Consequently, even with a high gain coefficient, the energy storage of the gain medium cannot be fully utilized. In order to solve this issue, the first-stage amplifier adopts the double-pass amplification to make full use of the energy storage in the gain medium. Additionally, a coupling lens, consisting of two lenses, is employed to efficiently couple the output signal beam into the crystal. To maintain control over beam quality post-amplification, selecting an appropriate beam overlap ratio becomes crucial. This factor represents the ratio of the signal beam's diameter to that of the pump beam.

All pump sources are fiber-coupled 808 nm laser diode modules with a maximum output power of 50 W and the laser linewidth (FWHM) is 4 nm (provided by DILAS). By adjusting the temperature of the cooling water, the central wavelength can be controlled to achieve effective matching with the absorption peak of the crystal. The fiber has a core diameter of 400 μm and a numerical aperture (NA) of 0.22. The pumping light is coupled into the crystal through the two end faces of the crystal by adjusting the coupling lens group. The waist size is optimized for output parameters in each amplifier stage based on experimental results.

Figure 2 shows the schematic diagram of pump light shaping and transmission. It is appropriate to have a beam overlap ratio of around 80~90% for the signal beam and pump beam, as it allows for the effective amplification of the signal beam power while maintaining good beam quality. Therefore, the design of the lens-coupling module adopts a variable focus structure, which allows for adjusting the pump beam diameter according to different beam overlap ratio requirements.

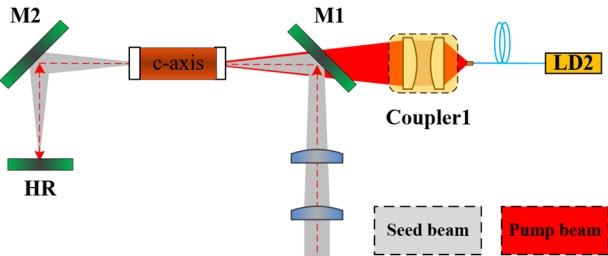

**Figure 2.** Illustration of the filling ratio between the seed beam and the signal beam.

According to the simulation results from Figure 3, the beam diameter of the pump light at a depth of 2 mm on the crystal end face is 400 μm for the first amplification module, with a pump power of 70 W. For the subsequent three stages, the pump light beam diameter is 600 μm, with a pump power of 90 W. A coupling lens group, consisting of two bonded lenses, is used to focus the beam into the crystal with beam-expansion ratios of 1:1 and 1:1.5, respectively. Finally, lenses L3 and L4 are used to expand and shape the amplified signal beam, ensuring that the beam passing through the last three amplifiers is nearly collimated and achieving an appropriate overlap ratio.

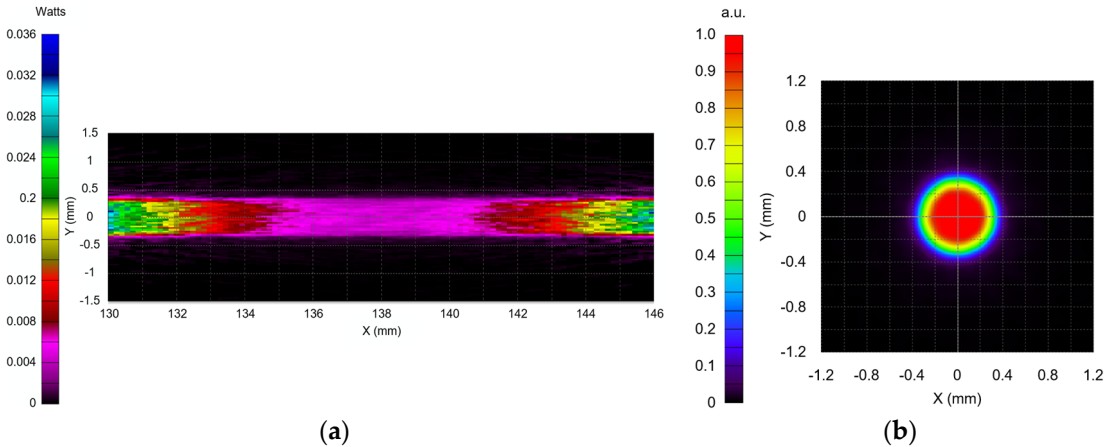

(**a**)                                                    (**b**)

**Figure 3.** (**a**) Longitudinal distribution of the optical field in the Nd:YVO$_4$ crystal and (**b**) Transverse distribution of the beam at the focal plane of the pump light.

### 2.2. Main-Amplifier Setup

The main amplification unit consists of a two-stage CW side-pumped Nd:YAG slab amplifier. The 0.6 at.% slab crystal (provided by Crylaser Inc.) adopts a ~60° cut angle parallelogram structure, with dimensions of 125 mm × 22 mm × 3.5 mm (width × thickness × length). Coating the surface of the slab with a layer of SiO$_2$ protective film can prevent light leakage caused by the sealing ring. Currently, optical silicones of good quality have refractive indices between 1.41 and 1.53, with industrial transparent silicones generally having higher refractive indices. Assuming the lowest refractive index of 1.41, the critical angle for total internal reflection of 1064 nm light from Nd:YAG material to transparent silicone is 50.86°. When the sealing ring or sealant is in close contact with the slab, it does not disrupt the original total internal reflection condition between YAG-SiO$_2$. The crystal is pumped from a single large surface, and the cooling is done through the same two large surfaces of the crystal. Figure 4 shows a schematic diagram of the laser transmission, pumping region, and cooling water distribution inside the slab amplification module. The laser enters the crystal slab from the center of the end face and then undergoes total internal reflection on the upper and lower surfaces of the slab, propagating along a zigzag path. Finally, it exits from the center of the opposite end face of the slab. The large surfaces of the slab crystal are coated with an evanescent wave protective film, which increases the damage threshold. Pumping light enters the crystal slab through one of its large surfaces and is reflected by the opposite surface, effectively being absorbed completely along the thickness direction of the crystal, as indicated by the yellow region in the diagram. The blue region represents the cooling water layer. Deionized water was used as the cooling fluid, with a water layer thickness of 2 mm, and cooled at a flow rate of 30 L/min at 20 °C.

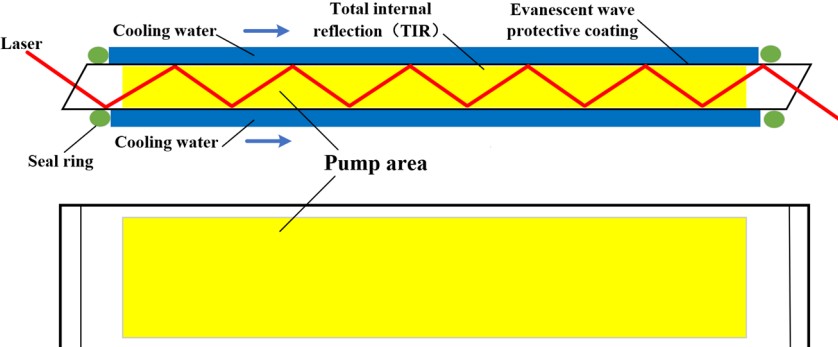

**Figure 4.** Schematic diagram of the laser beam path, pumping region, and cooling layer in the slab amplification module.

The slab amplifier module uses a horizontal pumping scheme, as shown in Figure 5. It consists of six vertically stacked arrays, with each array containing eight laser diode bars. Each bar has a power of 100 W (provided by Focuslight company, Xi'an, China), resulting in a maximum pumping power of 4800 W (nominal, with redundancy in reality). In the slow axis direction of the laser diode bar, the pump light is collimated by six cylindrical lens and overlaps on the crystal to form a uniformly intense optical spot. In the fast axis direction, the pump light passes through the cylindrical lens and enters a quartz waveguide. After multiple reflections on the upper and lower surfaces of the quartz waveguide, it finally combines inside the crystal to form a uniform optical spot.

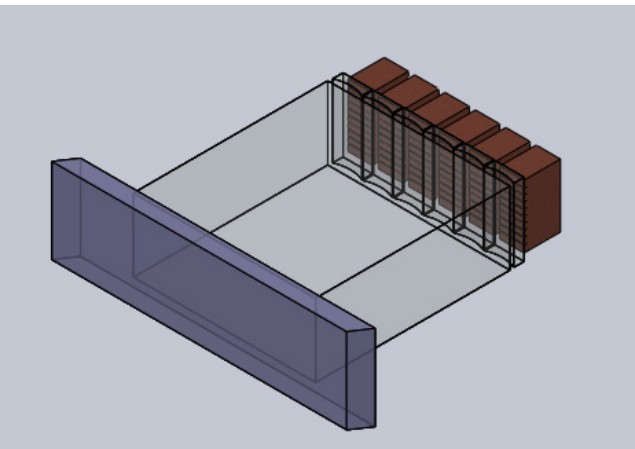

**Figure 5.** Schematic diagram of the side-pumping structure.

By optimizing parameters such as the spacing between laser diodes, the curvature of the cylindrical lens, and the length of the waveguide, based on simulation results, the uniformity of the pump light distribution on the crystal's cross-section exceeds 90%, as shown in Figures 6 and 7a. According to the simulation results, the spacing of the laser diode array used in our experiments is 1.7 mm; the curvature of the cylindrical lens is 15 mm, with a distance of 88 mm from the emitting surface of the laser diode; and the waveguide length is 80 mm. The dimensions of the pump region are 80 mm × 20 mm. The pump uniformity inside the slab crystal at different pump currents was tested using the image transfer method, as shown in Figure 7b.

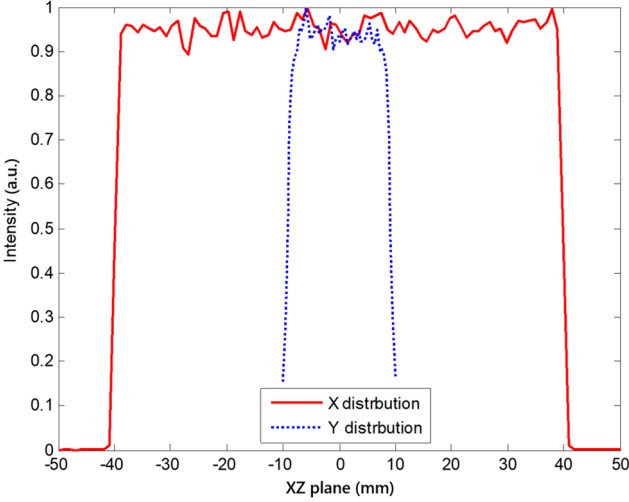

**Figure 6.** Transverse and longitudinal cross-sectional distribution of the pump light in the slab crystal.

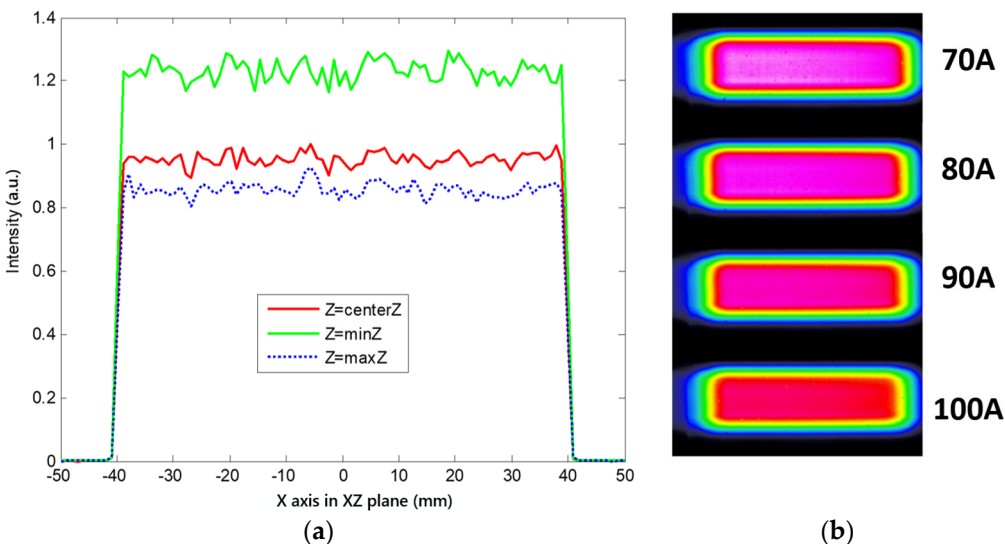

**Figure 7.** (**a**) Transverse cross-sectional distribution of the pump light on the front surface, center, and rear surface of the slab crystal and (**b**) Test results of pump light distribution inside slab crystal at different currents.

### 3. Experimental Results

Figure 8a shows the average laser power and extraction efficiency of each amplifier stage. The output power of the seed source is 6 mW. By adjusting the mode matching between the signal light and the pump light of the end-pumped amplifier, the seed light is finally amplified through the first-stage double-end-pumped Nd:YVO$_4$ preamplifier, resulting in an output power of 10.8 W. The total gain reaches 1800 times, as shown in Figure 8b. It is worth noting that such high gain makes it prone to self-oscillation between the crystal end face and the high reflectivity (HR) mirror. By adjusting the angle of the HR mirror to be tilted by ~3° with respect to the crystal end face, the self-oscillation light can be eliminated. Then, after three stages of preamplification, the output powers are 27.6 W, 51.8 W, and 80.5 W, respectively. The power extraction efficiencies of each preamplifier stage are as follows: 28.8%, 37.3%, 64.9%, 52.7%.

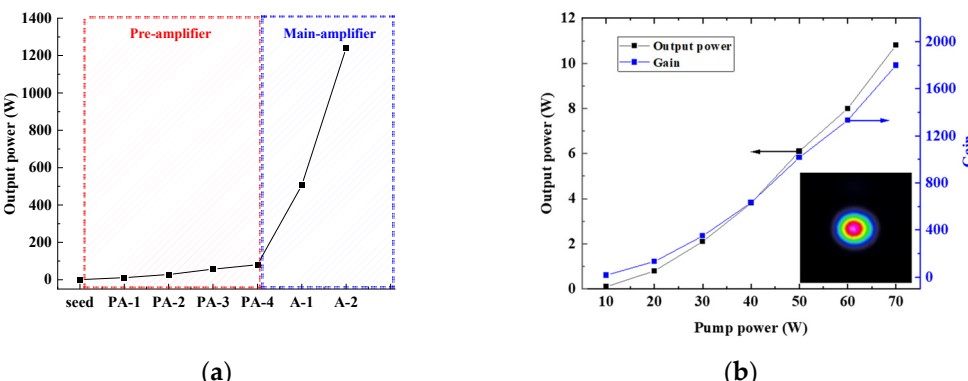

**Figure 8.** (**a**) The output power of each amplifier stage and (**b**) output power and gain of the first-stage pump amplifier.

The energy storage, gain, and depolarization test results of the continuous pump slab amplifier are shown in Figure 9. When the injection current is 100 A, the pump power reaches 5000 W, with an output power of 1500 W, corresponding to an optical-to-optical efficiency of 30%. The single-pass gain is 1.8, with a depolarization loss of 2.75%. Due to the depolarization loss and diffraction effects in the pre-amplification stage, the laser power entering the slab amplifier is limited to 75 W. Before entering the first-stage slab

amplifier, the laser beam is shaped into dimensions of approximately 3.5 mm × 20 mm. Between the two-stage slab amplifiers, there is a set of imaging transfer mirrors (L9 and L10) and a vacuum filter SF2 to suppress high-frequency components in the beam. After passing through the first and second stages of Nd:YAG slab amplifiers, the off-axis double-pass (incident angles were 2.8° and −1.4°, respectively) output powers are 506 W and 1240 W, with power extraction efficiencies of 28.4% and 48.9%, respectively. The maximum single-pulse energy is 62 mJ, with a repetition frequency of 20 kHz and a pulse width of 301 ns. The oscilloscope traces of the pulse series as well as the single pulse are illustrated in Figure 10a. The stability (RMS) of the output average power within 10 min is 0.83%. The beam quality diffraction limit of the laser output was estimated at 7 and 3, in slab width and thickness direction, respectively, due to severe thermal lensing; the far-field beam profile is shown in Figure 10b.

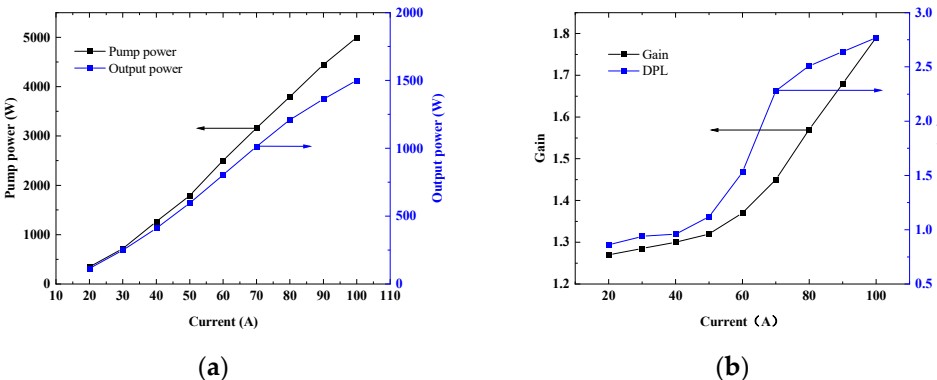

(a)  (b)

**Figure 9.** (**a**) Output power and (**b**) gain and bias current relationship curves for the slab amplifier module.

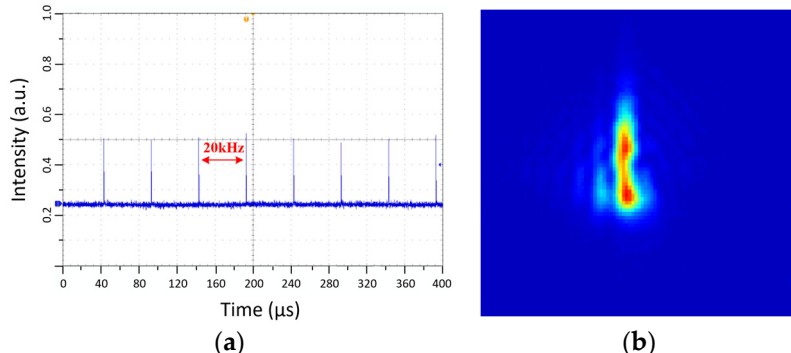

(a)  (b)

**Figure 10.** (**a**) The repetition frequency of the laser system output and (**b**) the far-field beam profile.

As the pulse width is 300 ns, the curves illustrating the variation of average output power and single-pulse energy at different repetition rates are depicted in Figure 11a. At a repetition rate of 1 kHz, the laser system yields an average power of 186 W, achieving a maximum single-pulse energy of 186 mJ. At 20 kHz, a maximum average output power of 1240 W is obtained. With the increase in repetition frequency, it is evident that the output power demonstrates a rising trend that gradually tends to stabilize. This can be explained by the fact that, as the pulse frequency increases, the pulse interval decreases, leading to a reduction in energy losses caused by spontaneous emission. The energy stored in the amplifier approaches a steady state. As the frequency continues to increase, pulse amplification eventually transitions into a steady-state amplification process. When the repetition rate is set to 20 kHz, the variation of average output power and single-pulse energy at different pulse widths is illustrated in Figure 11b. At a pulse width of 10 ns, the average output power is 880 W, with a single-pulse energy of 44 mJ. At 301 ns, the maximum single-pulse energy of 62 mJ is achieved.

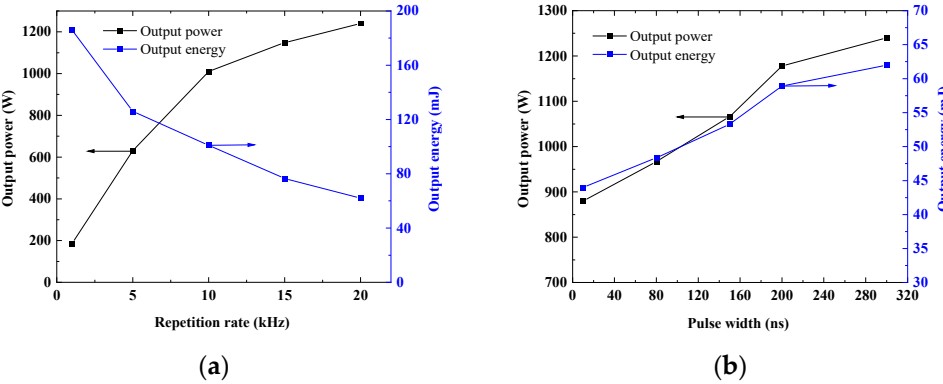

**Figure 11.** (**a**) Output power and pulse energy versus repetition rates and (**b**) output power and pulse energy versus pulse width.

After the laser pulse waveform from the seed source is amplified by the amplifier, the presence of gain-saturation effects causes the leading edge of the pulse waveform to become relatively steep, and the pulse width is correspondingly compressed and narrowed. To address this issue, we employ the pre-shaping of the seed source output waveform to compensate for the distortion of the pulse-leading edge caused by gain saturation. As a result, a relatively smooth pulse waveform output is obtained, as shown in Figure 12, with pulse widths of 10 ns and 301 ns (FWHM), according to the test results.

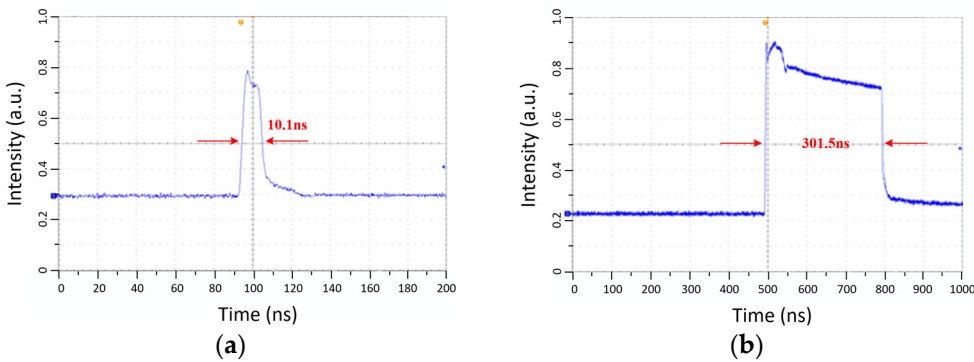

**Figure 12.** Waveform profile of the laser system output with a typical pulse width of (**a**) 10 ns and (**b**) 300 ns.

## 4. Discussion

Laser systems have been developed with higher average powers (20 kW) [22], shorter pulse widths (fs) [23], longer pulse widths (QCW) [22], larger single-pulse energies [24], improved self-oscillation suppression [25], and higher beam quality [26]. However, reference [22] reports a slab-based laser oscillator with an output power level of 20 kW, but a low repetition rate of only 400 Hz and a relatively wide pulse width of 200 μs. Reference [23] describes a slab-based MOPA laser system with a high repetition rate of 20 MHz, an average power of 1.1 kW, a pulse width of 615 fs, but a corresponding single-pulse energy of only 55 μJ. In reference [24], the laser system achieves a high single-pulse energy of 800 mJ at 1064 nm, but with a low repetition rate of 400 Hz. The parasitic oscillation suppression technique described in reference [25] requires complex preparation processes like bonding crystals to achieve good results. The method described in this paper achieves parasitic oscillation suppression by tilting the crystal angle during low-power amplification, a simpler and more feasible approach. Reference [26] reports a laser oscillator based on end-pumped Nd:YVO$_4$, achieving a beam quality factor M$^2$ of 1.3 with an output power of only 28.8 W, which is relatively easy to achieve in laser systems with low power output. Additionally, Comaskey et al. [27] reported a laser oscillator based on an LD array side-pumped slab amplifier, achieving output powers in the kilowatt range under continuous operation with

an optical–optical efficiency of 19%, while the slab amplifier we developed achieved an optical–optical efficiency of 30%.

At high repetition rates, achieving high-energy and narrow-pulse-width lasers simultaneously is a challenging research endeavor. Currently, there are two technical approaches to achieve this goal: bulk lasers based on the MOPA configuration [28,29] and fiber lasers [30]. However, the pulse waveform of conventional bulk lasers lacks flexible control, and fiber lasers operating at high repetition rates often exhibit limited single-pulse energy, typically not exceeding 10 mJ. The laser system we report achieves a single-pulse energy of 62 mJ at 20 kHz. Furthermore, the repetition rate, pulse width, and waveform are subject to flexible adjustments.

## 5. Conclusions

We have reported a kW level high-repetition-rate nanosecond laser, employing a structure of fiber, Nd:YVO$_4$, and Nd:YAG hybrid amplification. The pulse width and repetition frequency of the fiber seed source can be flexibly adjusted. By precisely matching the beam overlap ratio of the double-end-pumped Nd:YVO$_4$ pre-amplifier, we obtained a high-gain laser output. Finally, a dual-pass off-axis extraction scheme was used through the high-uniformity side-pumped Nd:YAG slab laser. The seed output was 6 mW at a repetition frequency of 20 kHz; we achieved an average power output of 1240 W with the total power extraction efficiency of 39.1% and a single-pulse energy of 62 mJ at the pulse width of 301 ns. Furthermore, the pre-compensation technique for the seed pulse waveform enabled flexible control of the laser temporal profile, making it highly promising for applications in laser cleaning. The final indicators are advanced in the current research field, offering the potential to bring new perspectives to the development of related areas.

**Author Contributions:** Conceptualization, H.L. and J.Q.; methodology, H.L., J.Q., Y.C. and X.S.; software, T.W.; validation, H.W., T.W. and Y.L.; formal analysis, H.W.; data curation, H.W.; writing—original draft preparation, H.L.; writing—review and editing, H.L.; supervision, Z.F.; project administration, Z.F.; funding acquisition, J.Q. and Z.F. All authors have read and agreed to the published version of the manuscript.

**Funding:** This work was supported by the Project: National Nature Science Foundation of China program (22227901).

**Institutional Review Board Statement:** Not applicable.

**Informed Consent Statement:** Not applicable.

**Data Availability Statement:** Data underlying the results presented in this paper are not publicly available at this time but may be obtained from the corresponding author upon reasonable request.

**Conflicts of Interest:** All authors declare no conflicts of interest.

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
