# Peer review of "1.2 kW, 20 kHz Nanosecond Nd:YAG Slab Laser System"

_photonics, doi:10.3390/photonics11040297_

Round 1
Reviewer 1 Report
Comments and Suggestions for Authors
Dear Authors,
I have deep respect to work which is beyond the paper, but I am not satisfied with the presentation of the work in the paper. I am lacking of some important details as well as some other details are not presented properly.
In the former part there is an overview of the field of slab lasers, mostly zigzag design where I am expecting to see a comparison of parameters available and what is special about presented laser setup since there are laser systems producing even higher average output (20 kW [1]), shorter (fs [2]) and longer pulse duration (qcw [1]), higher pulse energies [3], better parasitic oscillations suppression [4], as well as better beam quality [5]. I would like to see the circumscribe from the former works by Comaskey et al. [6] since they presented same design with similar output parameters already 30 years ago.
If you want to attract readers you should describe added values by your laser system in the abstract which actually suits better to a proceeding paper.
Follows the comments and suggestions to modify your paper that, I believe, would be valuable for potential readers. These comments are listed in the order of appearance in the paper.
In Figure 1, please increase the readability of captions.
In Figure 1 & 2, for PA1 it is indicated that beam waist is placed on the HR end mirror M0. Could you please comment whether this interprets the reality or is a consequence of an author arbitrariness.
In experimental setup section on lines 85-86, it is noted that 'splitting and isolation unit is consisted of half-wave plate (HWP) and a polarizing beam splitter (PBS).' The splitting and isolation will take effect only if the Faraday rotator (FR) would be involved. Since that I would recommend to mention the FR in the constituents of the splitting and isolation protection unit.
On the line 97, the word 'Translation: ' is unnecessary since the sentence followed is already translated?
On the line 104, the word 'fold' could be replaced with a meaningfully accurate 'angle'.
On the line 116, the value of stimulated emission cross-section should be either omitted or accompanied with the citation since it is not a general accepted information.
On the line 128, the phrase 'inverted particle population' is unappropriated and vague, please rephrase it to fit the physical meaning.
On the line 131, based on a definition given, the term 'beam filling factor' should be changed to appropriate 'beam overlap ratio'. Beam filling factor has different meaning. The term 'beam filling factor (ratio)' should be changed even in line 144, 148, and ongoing.
On lines 146-147, please explain the meaning of term 'dual adhesive variable focus structure,' or give the citation to this term, or just repair it to make sense.
In Figure 3, both charts are unreadable. Please increase the letter size to readable level.
In Figure 3a, please indicate the meaning of colors (add a color map indicating the field density).
In Figure 3b, please add a colormap. The chart is meaningless without the colormap.
In the line 190, there is phrase 'cross-section can be improved to exceed 90 %'. Does this means that the optimization was just modeled without experimental verification? I would suggest to simplify the whole paragraph starting on the line 188, stating the real pumping scheme used in the experiment, without giving too many details on the pumping optimization and write a separate article in which a deeper description of the pumping optimization will be present.
In figure 6 & 7, the units for horizontal axis are missing.
On the line 206, there is sentence 'At this point, it is necessary to ensure that the incident signal light is also tilted at the same angle,' which I believe is not valid since by tilting the end mirror M0 (mentioned before) the alignment of PA1 and the entrance part (including the isolation unit) is involved. I would recommend to reframe or to omit this part.
On lines 242-243, there is claim 'we obtained a high-gain and high-beam-quality laser output,' which does not match with the beam quality evaluation on the line 219. I would suggest to don't comment the beam quality.
For all figures placed in the paper, Chinese (most probably) captions appear when mouse cursor is placed over.
To give readers a complex figure of the laser system output parameters, I would show the output beam profile whatever it is.
Literature
1. Y. Guo et al. 24.6 kW near diffraction limit quasi-continuous-wave Nd:YAG slab laser based on a stable–unstable hybrid cavity (2020).
2. P. Russbueldt, et al. Compact diode-pumped 1.1 kW Yb:YAG Innoslab femtosecond amplifier (2010).
3. Shiguang Li et al., Laser-diode-pumped zigzag slab Nd:YAG master oscillator power amplifier (2013).
4. Arun Kumar Sridharan et al. Zigzag slabs for solid-state laser amplifiers: batch fabrication and parasitic oscillation suppression (2006).
5. Zhe Ma et al. Monolithic Nd:YVO4 slab oscillator–amplifier (2007).
6. B. J. Comaskey et al. One-kilowatt average-power diode-pumped Nd:YAG folded zigzag slab laser (1993).
Comments on the Quality of English Language
The quality of English is acceptable.
Reviewer 2 Report
Comments and Suggestions for Authors
This paper studied and developed a kW-level high-repetition-rate nanosecond laser, employing a structure of fiber, Nd:YVO4, and Nd:YAG hybrid amplification. The final indicators are advanced in the current research field, offering the potential to bring new perspectives to the development of related areas.
Here are some suggestions for modifications to this article:
1. This paper mainly introduces the laser output parameters at 300ns/20kHz. Why choose the parameters at 300ns/20kHz? If used for laser cleaning, narrow pulse width and high peak power may be more advantageous. Please provide appropriate explanations. In addition, as the feature of this paper is the adjustability of the pulse width and repetition frequency , it is recommended to provide the relationship between the pulse energy and the variation of pulse width and repetition frequency.
2. What is the extraction efficiency of different amplification levels during laser operation? How is the extraction efficiency of different amplification stages balanced during laser design? Suggest the authors provide a brief explanation.
3. Some images in this paper need to be modified or improved in quality. 1) Please remove the outer frames of Figure 10 and Figure 11. 2) Adjust the axis font in Figure 8(a) to match the others. 3) The quality and clarity of Figure 1 and Figure 3 need to be improved.
4. Please enhance the conclusion section by suggesting to include a more comprehensive summary that highlights the innovative aspects of the research findings.
Comments on the Quality of English LanguageThere are some grammar mistakes in current version. English should be polished carefully.
Reviewer 3 Report
Comments and Suggestions for Authors
This manuscript is about the high-energy power laser amplifier based on both Nd:YVO4 end-pumped structure and Nd:YAG slab structure. The authors should answer the following questions before the manuscript can be accepted.
1. As the power can research ~1 kW in the Nd:YAG-slab amplifiers, the side pump structure is the key in determining the final performance of the amplifier while more details should be given on the side-pump system and a trace simulation of the pump lighted is preferred.
2. Is the cooling liquid regular water or something else?
3. The vendors of the crystals and pump sources should be provided.
Round 2
Reviewer 1 Report
Comments and Suggestions for Authors
Dear Authors,
Thank you for the effort you gave to the revision of the paper, I really appreciate it very much and I believe that even the audience would appraise it.
I have suggested the paper to accept with minor revision since I believe the discussion part in the present form is really unnecessary. My concern was to enhance the introduction part and see the motivation for selecting right the parameters of your system you selected (since there are many similar laser system designs beyond the kW-class systems). Just simple clause that such combination of energy and reprate gives no compromise to ablation efficiency and speed of processing would be enough. Even though, fs lasers are now under demand for laser cleaning [1]. I believe that only second paragraph (starting in line 297) is relevant to the issue I wanted to point out. I will let it your choice whether to leave the whole discussion paragraph as it is, or somehow redistribute the text to the introduction part, where these statements suits more.
Sorry for the complication, this problem probably arise because of my not very specific comment on that.
1: Maharjan, N., Zhou, W., Zhou, Y., & Guan, Y. Femtosecond laser cleaning for aerospace manufacturing and remanufacturing.
doi: 10.1109/CLEOPR.2017.8119087 (2017).
